# NFκB Inhibition Mitigates Serum Amyloid A-Induced Pro-Atherogenic Responses in Endothelial Cells and Leukocyte Adhesion and Adverse Changes to Endothelium Function in Isolated Aorta

**DOI:** 10.3390/ijms20010105

**Published:** 2018-12-28

**Authors:** Abigail Vallejo, Belal Chami, Joanne M. Dennis, Martin Simone, Gulfam Ahmad, Adrian I. Abdo, Arpeeta Sharma, Waled A. Shihata, Nathan Martin, Jaye P. F. Chin-Dusting, Judy B. de Haan, Paul K. Witting

**Affiliations:** 1Discipline of Pathology, Sydney Medical School, The University of Sydney, Camperdown, NSW 2006, Australia; aval3972@uni.sydney.edu.au (A.V.); belal.chami@sydney.edu.au (B.C.); joanne.dennis@sydney.edu.au (J.M.D.); msim6013@uni.sydney.edu.au (M.S.); gulfam.ahmad@sydney.edu.au (G.A.); nmar7397@uni.sydney.edu.au (N.M.); 2Heart Research Institute, Newton, NSW 2053, Australia; adrian.abdo@hri.org.au; 3Baker Heart and Diabetes Institute, Victoria 3004, Australia; Arpeeta.sharma@baker.edu.au (A.S.); waled.shihata@monash.com.au (W.A.S.); jaye.chin-dusting@baker.edu.au (J.P.F.C.-D.); judy.dehaan@baker.edu.au (J.B.d.H.); 4Department of Medicine, Monash University, Victoria 3500, Australia; 5Cardiovascular Disease Program, Biomedicine Discovery Institute, Monash University £Department of Pharmacology, Monash University, Victoria 3800, Australia; 6Department of Immunology, Monash University, Victoria 3004, Australia; 7Department of Physiology, Anatomy & Microbiology, School of Life Sciences, La Trobe University, Bundoora, VIC 3083, Australia

**Keywords:** nuclear, transcription, endothelium, atherosclerosis, serum amyloid A, aorta

## Abstract

The acute phase protein serum amyloid A (SAA) is associated with endothelial dysfunction and early-stage atherogenesis. Stimulation of vascular cells with SAA increases gene expression of pro-inflammation cytokines and tissue factor (TF). Activation of the transcription factor, nuclear factor kappa-B (NFκB), may be central to SAA-mediated endothelial cell inflammation, dysfunction and pro-thrombotic responses, while targeting NFκB with a pharmacologic inhibitor, BAY11-7082, may mitigate SAA activity. Human carotid artery endothelial cells (HCtAEC) were pre-incubated (1.5 h) with 10 μM BAY11-7082 or vehicle (control) followed by SAA (10 μg/mL; 4.5 h). Under these conditions gene expression for TF and Tumor Necrosis Factor (TNF) increased in SAA-treated HCtAEC and pre-treatment with BAY11-7082 significantly (TNF) and marginally (TF) reduced mRNA expression. Intracellular TNF and interleukin 6 (IL-6) protein also increased in HCtAEC supplemented with SAA and this expression was inhibited by BAY11-7082. Supplemented BAY11-7082 also significantly decreased SAA-mediated leukocyte adhesion to apolipoprotein E-deficient mouse aorta in *ex*
*vivo* vascular flow studies. In vascular function studies, isolated aortic rings pre-treated with BAY11-7082 prior to incubation with SAA showed improved endothelium-dependent vasorelaxation and increased vascular cyclic guanosine monophosphate (cGMP) content. Together these data suggest that inhibition of NFκB activation may protect endothelial function by inhibiting the pro-inflammatory and pro-thrombotic activities of SAA.

## 1. Introduction

The endothelium plays a critical role in regulating immune processes in the vasculature by providing a structural barrier, a non-thrombogenic surface and maintaining vascular tone [1]. However, pathophysiological activation of the endothelium facilitates vascular inflammation and endothelium dysfunction is implicated in the progression of atherosclerosis and other chronic inflammatory diseases [2]. Endothelial cell activation occurs in the earliest stages of cardiovascular disease and is linked to loss of barrier function, focal inflammation, lipid accumulation and pro-coagulation [3]. Endothelium activation and dysfunction also impact on cardiovascular risk factors such as hypertension [4], type 1 diabetes [5] and rheumatic disorders [6] all diseases characterized by a sustained increase in circulatory levels of several inflammatory biomarkers including serum amyloid A (SAA).

SAA is an acute phase protein that is significantly up-regulated by infection or injury [7]. Under these conditions (cardiovascular diseases, type 1 diabetes, rheumatic disorders) SAA is produced primarily in the liver [8] although other source include coronary artery endothelial cells [9]. Whilst SAA is an acute phase protein, circulating levels are evident in various inflammatory disorders such as non-insulin dependent diabetes and can exceed 2 mg/L in chronic inflammatory conditions [10].

In addition to functioning as an inflammatory biomarker, SAA is proposed to stimulate several cellular activities linked to atherogenesis. Plasma SAA predicts adverse effects in patients with vascular disease and SAA is found at sites of thrombus and plaque rupture in atheroma [11,12]. SAA is also known to induce pro-inflammatory and pro-thrombotic activities in endothelial cells, including up-regulation of adhesion molecules, decreased nitric oxide (NO) production and bioactivity and accumulation of reactive oxygen species (ROS) [13], all effects linked to endothelial dysfunction, which precedes atherogenesis [2,14]. Furthermore, SAA induces production of tissue factor (TF) and tumour necrosis factor (TNF) in peripheral blood mononuclear cells (PBMC) and immortalised macrophage cells [15,16] as well as other pro-inflammatory cytokines such as IL-1B, monocyte chemoattractant protein-1 (MCP-1), IL-6, IL-8 and macrophage inflammatory protein-1 alpha (MIP-1α) in both monocytes and local stromal cells [15,17].

*In vivo* studies also indicate a pro-atherogenic role for SAA, specifically, the acceleration of atherosclerosis in apolipoprotein E-deficient mice via promotion of recruitment and adhesion of macrophages at the inflammatory sites [18,19]. The pro-inflammatory mediators (MCP-1, MMP-9, TF) and vascular adhesion molecules (ICAM-1, VCAM-1) are increased upon SAA stimulation in atherosclerotic lesions [18]. Moreover, SAA up regulates the low-density lipoprotein (LDL) scavenger receptor that drives foam cell and lesion formation [20].

The pro-atherogenic effects of SAA may be mediated via the transcription factor NFκB—a key gene expression regulator of various inflammatory cytokines, chemokines and adhesion molecules [21]. NFκB can be induced by SAA in endothelial cells [22,23] and is implicated in SAA-mediated TF and/or cytokine expression in monocytes [15,17,24], macrophages [16] and endothelial cells [25]. Activation of cell surface receptors for SAA such as receptor for advanced glycation end products (RAGE), toll-like receptor (TLR) and formyl peptide receptor-like 1 (FPRL-1) also stimulate NFκB activation [22] and inhibition of these receptors, or SAA activity, can ameliorate SAA-mediated endothelial dysfunction [26]. The cytokine TNF is up-regulated by SAA in many cell types and stimulates adhesion molecule expression in endothelial cells, also induces nuclear factor kappa-light-chain-enhancer of activated B cells-1 (NFκB1) a member of the NFκB family of transcription factors [27]. Overall, these data suggest that NFκB may play a pivotal role in mediating the pro-atherogenic effects of SAA in the endothelium under pathophysiological conditions. The present work employs ex vivo and in vivo studies aimed to determine whether pharmacologic inhibition of NFκB activation and down-regulation of its transcriptionally activated target genes protects the endothelium from SAA induced adverse effects.

## 2. Results

Consistent with previous data indicating that SAA increased VEGF expression in primary endothelial cells [22], cultured HCtAEC supplemented with 10 μg/mL SAA showed ~2-fold increase in secretory levels of VEGF relative to control cells (Figure 1), albeit this did not reach statistical significance. BAY11-7082 (1–100 μM) dose-dependently inhibited the level of secretory VEGF induced by SAA (Figure 1). At 10 μM (BAY11-7082), secretory VEGF was reduced by ~40% compared to SAA alone and this concentration of the NFκB inhibitor (BAY11-7082) was selected for all further studies on gene and protein analysis. This selected (active) dose of BAY11-7082 (10 μM) is lower than that used by others [28,29] and provides an advantage of minimising potential non-specific inhibitory activity in cultured HCtAEC.

### 2.1. Gene Expression

SAA-stimulation of TF and TNFα gene expression was selected for assessment as both proteins play a role in SAA-mediated endothelial activation [22,24]. Specifically, TF expression is induced by inflammatory cytokines in atherosclerotic plaques and acts as a physiological trigger of thrombus formation through potent activation of the coagulation cascade at the site of plaque rupture and; TNFα targets vascular endothelium and increases the expression of several other pro-inflammatory, pro-coagulant and pro-apoptotic genes [30] in a cellular feed-back loop. This also reduces NO bio-availability [31,32] secondary to NO scavenging by reactive oxygen species (ROS) or inhibition/decline in NO biosynthesis [33], which potentiates vascular endothelial dysfunction. HCtAEC supplemented with SAA at normo-glycaemic conditions (media containing 5 mM D-Glucose), showed significantly higher TNF mRNA (>30x; *p* < 0.05 vs. control cells) and discernibly higher but non-significant TF mRNA expression (Figure 2). Pre-treatment of HCtAEC with BAY11-7082 (10 μM) prior to SAA supplementation significantly inhibited the TNF mRNA expression and showed marked but non-significant inhibition of TF mRNA expression in the same cultured endothelial cells (Figure 2) suggesting that the selected dose of inhibitor was able to inhibit SAA pro-inflammatory actions on HCtAEC.

### 2.2. Inflammatory Protein Expression

Enhanced expression of TNF is linked to the downstream production of secretory IL-6 as marker of bioactivity for this cytokine. For example, TNF alpha has been demonstrated to elicit expression of IL-6 at gene and protein levels in rat intestinal epithelial cells [34]. In our hands, cell confluency-normalised secretory IL-6 protein increased (~10-fold; *p* < 0.05) in HCtAEC exposed to 10 μg/mL of pro-inflammatory SAA. By contrast, pre-treatment of cells with BAY11-7082 markedly abated SAA-mediated secretion of IL-6 yielding levels below that detected in the corresponding vehicle-treated control cells (Figure 3). Taken together, these data suggest that pharmacological inhibition of NFκB can diminish downstream inflammatory markers.

### 2.3. Immunocytochemistry Analysis

Next, immunocytochemistry experiments were performed to assess the level and distribution of IL-6 and TNF (Figure 4) in order to determine whether gene regulation resulted in corresponding transcriptional regulation of the inflammatory cytokines. HCtAEC treated with 10 μg/mL SAA displayed more IL-6 positive (red) staining in the cell cytoplasm than the control group in the absence of SAA stimulation, (compare Figure 4A,B). IL-6 immuno-positive staining markedly diminished in cells pre-treated with 10 μM BAY11-7082 prior to SAA insult (compare Figure 4B,C and corresponding insets highlighting IL-6^+^ immune-fluorescence within single cells). As predicted based on SAA-mediated increases in TNF mRNA (see Figure 2), immuno-positive TNF increased in the cytoplasm of HCtAEC following incubation with 10 μg/mL of SAA (compare Figure 4D,E) while a decreased cytoplasmic accumulation of TNF was noted in cells pre-incubated with BAY11-7082 (Figure 4F). These results confirm that BAY11-7082-mediated inhibition of NFκB concomitantly decreases secretory IL-6 protein (consistent with data shown in Figure 3) while also demonstrating that TNF protein expression was also sensitive to inhibition of NFκB in HCtAEC in the presence of SAA.

### 2.4. Leukocyte Endothelial Adhesion 

Aortae isolated from ApoE-deficient mice administered with SAA showed a significant increase in adherent leukocytes (Figure 5A,C), which was almost completely inhibited by pre-treatment with BAY11-7082 prior to administering SAA (Figure 5B,C). Under these experimental conditions, control mice that received sterile phosphate buffered saline (as a control) showed minimal adherent cells that did not differ markedly from the aortae from mice treated with BAY11-7082, suggesting that NFκB inhibition effectively decreased endothelial surface adhesion elicited by SAA (refer to Figure 5A,D).

Interestingly, vascular cellular adhesion molecule-1 (VCAM-1) promotes adhesion of monocytes to activated endothelium and its expression on the endothelial cell surface is regulated via phophorylative activation of mitogen activated protein kinases (MAPK) through a mechanism that involves NFκB and TNF [35], a mechanism common to other cell types [36]. Therefore, we next investigated the levels of ERK1/2 activation in homogenates of aortae from mice treated with vehicle (control), SAA or combined BAY11-7082/SAA. Aortae from mice treated with SAA alone showed a significant increase in total phosphorylated extracellular signal-regulated kinases (p-ERK1/2) protein levels, which was markedly inhibited by BAY11-7082 supplementation to baseline levels (Figure 6). Together these data suggest that SAA stimulates leukocyte endothelial adhesion to vascular endothelium via NFκB/TNF-dependent pathways, which can be significantly inhibited by pre-treatment with the NFκB inhibitor BAY11-7082.

### 2.5. Vascular Function Test

Addition of SAA to isolated rat aortic rings inhibited endothelium-dependent relaxation in the presence of ACh (Figure 7A); this loss in vaso-relaxation has been ascribed previously to SAA stimulating increased production of superoxide radical anion that deactivates vaso-dilating nitric oxide [13]. Pre-incubation of aortic segments with BAY11-7082 significantly improved endothelium-dependent relaxation in the presence of SAA compared to rings without BAY11-7082 pre-treatment. However, the level of vascular relaxation did not reach that of the control (in the absence of SAA and BAY11-7082) and was also markedly less than endothelium-independent relaxation stimulated by the endothelium-independent vaso-dilator, SNP (Figure 7A).

Owing to increased TNF expression after SAA insult and its impact on reducing the bioavailibity of NO implicated in the vascular relaxation, we measured cGMP levels as an indicator of endothelium function. The levels of aortic cGMP, that are related to the presence and bioactivity of NO were consistent with the altered relaxation profiles exhibited in the same aortic tissue segments (Figure 7B). Thus, aortic segments stimulated with SAA showed significantly reduced tissue cGMP compared to controls in the absence of SAA activation. Furthermore, pre-treatment with BAY11-7082 prior to incubation with SAA significantly increased aortic cGMP compared to vessel segments incubated with SAA alone, although the extent of cGMP recovery did not reach control or SNP-elicited levels (Figure 7B).

## 3. Discussion

Previous studies have shown SAA to be a significant promotor of endothelial dysfunction, a precursor to the development of vascular disease [7,26]. Attempts to use pharmacological approaches to block SAA receptor activation have proven to be less effective in ameliorating SAA bioactivity compared with athero-protective HDL (a potential *in vivo* regulator of SAA activity) [23]. Herein, the data demonstrate that inhibition of the transcription factor NFκB, concomitantly inhibits the pro-inflammatory and pro-thrombotic activity of SAA on cultured vascular endothelial cells Thus, HCtAEC stimulated by SAA induced the expression of pro-inflammatory mediators including IL-6 and TNF while prothrombotic TF and growth factor VEGF also increased in the presence of SAA. Pre-treatment of the endothelial cells with the NFκB inhibitor BAY11-7082 blocked SAA-stimulated mRNA expression of TNF and TF in the endothelium. Likewise, IL-6 and TNF proteins decreased in the presence of the NFκB inhibitor post-stimulation with SAA. In aortae isolated from apolipoprotein E-deficient mice, supplementation of the mice with BAY11-7082 also significantly decreased SAA-mediated leukocyte adhesion. Further, freshly isolated rat aortic rings bathed in media containing BAY11-7082 both reversed SAA-mediated inhibition of vessel relaxation stimulated by acetylcholine and increased aortic cGMP levels in parallel indicating markedly improved endothelium-dependent vasodilatation. Overall, these data suggest that blockade of this transcription factor may effectively inhibit SAA bioactivity and ameliorate vascular inflammation that may be central to many developing vascular pathologies.

In studies with cultured human carotid artery endothelial cells performed here, addition of recombinant SAA up-regulated inflammation through promoting cellular production of several mediators including TNF and IL-6 as well as pro-thrombotic TF, consistent with prior work [7,22]. These pro-inflammatory and pro-thrombotic elements are implicated in atherosclerosis and other chronic inflammatory diseases. Cell-derived mediators such as TNF are critical to immune regulatory and pro-inflammatory responses that are central to combating acute and chronic events. The data herein indicate that TNF mRNA increases after exposure of endothelial cells to SAA, which is in agreement with previous studies that observed a similar TNF response in other primary endothelial cells [22] as well as human monocyte cell line THP-1 [7]. Indeed, increased production of TNF has been linked to a number of pathologies where SAA is also increased [7]. In particular, TNF levels have been found to be significantly up regulated in mice prone to developing atherosclerosis and levels of TNF were directly linked to SAA increase SAA plasma [18]. 

Furthermore, the chemokine TNF activates cell surface expression of important adhesion molecules (ICAM-1, VCAM-1) that are central to leukocyte recruitment [37]. In addition, TNF acts as an instigator of the expression and biosynthesis of other mediators of the innate immune system such as TF and IL-6 [38]. This is particularly relevant in the model of SAA-mediated endothelial dysfunction used here where increased TF and IL-6 were detected, consistent with TNF amplifying the local pro-inflammatory environment and promoting endothelial dysfunction, a characteristic of the SAA-stimulated vascular endothelium [13]. In aortic homogenates, significant elevation in SAA induced ERK1/2 activation and subsequent inhibition by BAY11-7082 treatment suggest that NFkB activation stimulates ERK1/2 expression (likely involving TNF [35,36] and conversely pharmacologic NFkB inhibition down regulates ERK1/2 activation, which then represents the mechanism to explain decreased leukocyte adhesion to the vascular endothelium observed in *ex vivo* studies with isolated aortae performed here. This outcome is completely consistent with other studies that report TNF-α induced activation of MAPKs leading to inflammatory response in endothelial [35] and human proximal tubular epithelia cells [39], which can be inhibited by selective inhibitors including BAY11-7082.

Closely related to an enhanced pro-inflammatory state, TF and its cascade of pro-coagulatory events is through interaction with thrombin [40]. Thus, TF is clinically significant in thrombus formation and is implicated in atherogenesis [41] and can be a predictor of acute coronary syndromes [42]. Discernibly higher TF mRNA was detected in HCtAEC stimulated with SAA consistent with other cell models of SAA induced endothelial dysfunction [22,24]. Interleukins and particularly, IL-6 may play a multi-functional role in SAA-mediated endothelial dysfunction [7]. Herein, this study, IL-6 was a focus as its relevance to atherosclerosis were several fold, thus IL-6: (i) plays primary role in hepatic C-reactive protein synthesis which leads to generation of pro-inflammatory cytokines [43]; (ii) possess significant pro-coagulant activity [44] that may augment SAA-stimulation of TF; (iii) activates endothelial cells which in turn express cellular adhesion molecules (ICAM, VCAM) leading to plaque formation [19,45] and (iv) has some potential to develop atherosclerosis in mice when administered (IL-6) exogenously [46] and therefore, may play a role in SAA-mediated pro-atherogeneic activity. Increased accumulation of IL-6 has been shown by others to promote a local pro-inflammatory environment [45] as well as increase vascular permeability to blood borne components [47]. In this study, we observed a marked increase in IL-6 protein secretion by SAA-stimulated HCtAEC. Interestingly, secretion of IL-6 is not only implicated in downstream activities of SAA but may also act upstream of the acute phase protein. Thus, IL-6 is known to be a potent inducer of cellular synthesis of SAA during inflammation [48] by various cell types including those of the liver such as hepatocytes, Kupffer cells [8] and in human monocytic leukaemia cell lines (THP-1) [49]. If SAA-stimulation leads to production of IL-6 by a dysfunctional endothelium, then this interleukin may induce the production of additional SAA, thereby establishing an autocrine loop and effectively exacerbating endothelial dysfunction. Evidence for this IL-6-driven autocrine loop has been reported previously in thrombin-activated endothelial cells [50].

It should be noted that for SAA-induced IL-6 production by HCtAEC, the secretory form of IL-6 was more prevalent than intracellular accumulation of this interleukin, suggesting that a majority of IL-6 produced is trafficked to the extracellular domain. Similar cytokine trafficking trends have been found to occur in studies of rheumatoid arthritis patients [51]. The observation that much of the interleukin was exocytosed 24 h after SAA insult supports the notion that trafficking of this cytokine favours its release into the extracellular compartment and this is likely to involve a mechanism of exocytosis of vesicles enriched with IL-6 as detected by immune-fluorescent microscopy in this study (refer to Figure 4).

Of the observed mediators in our system, IL-6 displayed the strongest response to BAY11-7082 suggesting that the NFκB transduction pathway is central in its induction by SAA. Following the premise that IL-6 production is intensified by autocrine regulation, the same may hold true for its inhibition or depletion. In related studies involving rheumatoid arthritic synovium, SAA showed a similar ability to up-regulate IL-6 and this bioactivity was markedly abrogated by NFκB inhibition using BAY11-7082 [52]. 

In addition, previous studies in endothelial cells and synoviocytes have reported a role for SAA in the promotion of angiogenesis through its ability to induce VEGF [22] and this pro-angiogenic activity can be partially inhibited by pharmacological blockade of a range of SAA receptors at the cell surface [22]. The present results showing that BAY11-7082 dose-dependently decreased VEGF production in endothelial cells stimulated with SAA also indicate that inhibition of the canonical NFκB pathway is an effective alternative to pharmacological approaches aimed at blocking SAA receptors on the endothelial cell surface. Whilst, blockade of NFκB resulting in a corresponding decline in angiogenic VEGF has not been well documented in endothelial cell models, related studies with breast cancer cell lines such as those in MDA-MB-231 have displayed successful VEGF suppression with an NFκB inhibitor [53].

Owing to high rate of atherosclerosis and associated cardiac complications resulting from plaque formation and rupture; we aimed to investigate the potential role of BAY 11-7082 on endothelial adhesion and vascular functions. Our results showed significant reduction of the endothelial adhesion in mice treated with NFκB inhibitor compared to SAA alone group. These findings suggest that BAY11-7082 prevents the activation of NFκB thus inhibiting the expression of adhesion molecules and minimising the risk of vascular plaque formation. Further to endothelial adhesion implicated in plaque formation is the lost vascular elasticity occurring under inflammation mediated oxidative stress leading to depletion of nitroxides essential to vascular dilatation and hemodynamics. Endothelial cell dysfunction can lead to increased permeability to lipoproteins, leukocytes adhesion and cytokines generation [54]. To assess the role of BAY11-7082 on vasoconstriction and vasodilatation we treated the rat aortae with SAA alone and in supplementation with BAY11-7082. The vasodilatation was significantly improved in the aortas pre-treated with BAY11-7082 compared to SAA alone. These findings were further supported by assessments of aortic cGMP levels, which were significantly higher in vessel segments treated with BAY 11-7082 suggesting its potential to limit SAA-mediated decrease in nitric oxide production/bioactivity on the vascular endothelium.

## 4. Limitations for the Study

One clear limitation to this study is the mode of delivery of SAA to the endothelial cells in the absence of serum. SAA is an apolipoprotein for HDL and avidly binds to this lipoprotein and in the process displace apolipoprotein AI [55,56,57]. SAA bioactivity may be regulated by binding to high-density lipoprotein (HDL). We have previously shown that this binding of SAA to HDL becomes saturated at approximately 5 mol SAA per mol HDL [23]. Therefore, for SAA to elicit a biological response on endothelial cells levels of SAA should be in significant excess to circulating HDL (for example, when the mol ratio SAA:HDL is > 5). Such conditions may be evident for chronic inflammatory disorders such as diabetes and rheumatoid arthritis [58,59] where SAA levels are chronically elevated and natural variation amongst humans can see lower levels of HDL that predispose to vascular inflammation [60] and thereby, a high SAA to HDL ratio may compromise HDL anti-inflammatory activity to promote atherosclerosis [61].

## 5. Conclusions

In summary, our results confirm SAA induced over expression of TNF and IL-6; increased leukocytes endothelial adhesion and compromised vascular activity. Pre-treatment with BAY11-7082 effectively reversed these changes supporting a role of NFκB in SAA bioactivity. Moreover, NFκB inhibition by BAY11-7082 may be a potential mechanism to modulate thrombus formation in the SAA-stimulated endothelium, a risk factor of cardiac disease particularly in diabetic patients. 

## 6. Materials and Methods

### 6.1. Materials

Chemicals, cell culture reagents and proteins were of the highest quality grade available and sourced from Sigma-Aldrich, Australia unless specified otherwise.

### 6.2. Cell Culture

Commercial human carotid artery endothelial cells (HCtAEC) (Cell applications Inc, San Diego, CA, USA) were cultured in MesoEndo Cell Growth Medium (Cell applications, California, USA) containing endothelial cell growth serum (ECGS) (Millipore, Sydney, Australia). Cells were grown to ~90% confluence at 37 °C in a humidified atmosphere with 5% (*v*/*v*) CO_2(g)_ as described previously [23]. Each in vitro test was performed on HCtAEC sub-cultured to passage 4.

### 6.3. Dose Selection for In Vitro Cell Culture Experiments with BAY11-7082

In our previous reports [17,22] 10 μg/mL recombinant SAA (Human Apo-SAA from Peprotech (Rocky Hill, NJ, USA). Catalogue Number: 300-13 PeproTech, has induced significant expression of TF, TNF and VEGF in cultured HCtAEC exposed to SAA and was therefore used (10 μg/mL) in all current *in vitro* experiments conducted in this study. This SAA concentration represents a median value between normal, low circulating and acute phase SAA levels (5–25 μg/mL; [17]). We routinely and exhaustively test preparations of recombinant SAA for LPS contamination and determined that batches employed here contained <2 pg LPS/μg SAA/mL. This low-level contamination was unable to induce pro-inflammatory/pro-coagulant responses in human peripheral blood monocytes that are highly sensitive to LPS and denaturing the SAA protein by heating to 100 °C ablates it biological activity on isolated peripheral blood monocytes while similar treatment of LPS has no impact on its biological action [40]. All reagents and media were rigorously tested for endotoxin levels using the Limulus Amoebocyte Lysate (LAL) buffer and endotoxin standards, visualised with Spectrozyme LAL (American Diagnostica, Stamford, CA, USA). Reagents were discarded if endotoxin levels were >5 pg/mL.

Owing to positive association of VEGF-A with atherosclerosis particularly in ApoE-/- mice [62,63] optimal dose of the NFκB inhibitor BAY11-7082 was selected based on its inhibitory effects on VEGF measured by an enzyme-linked immunosorbent assay (ELISA) as described previously [23]. Briefly, cells (density 1 × 10^7^ cells/well) were designated to four different treatment groups and pre-treated with vehicle (dimethyl sulfoxide; DMSO as a control) or 1, 10 or 100 μM BAY11-7082 in DMSO. Control and drug-treated cells were then incubated in HEPES Physiological Saline Solution (HPSS) (containing: 22 mM HEPES, 124 mM NaCl, 5.6 mM D-Glucose, 5 mM NaHCO_3_, 5mM KCl, 1.5 mM CaCl_2_, 1 mM MgCl_2_, 0.16 mM Na_2_HPO_4_, 0.4 mM NaH_2_PO_4_, adjusted to pH 7.4) at 37 °C and 5% CO_2(g)_ for 1.5 h. Cells were then treated with 10 μg/mL recombinant human Apo-Serum Amyloid A (SAA1) (Peprotech, USA) and incubated at 37 °C and 5% (*v*/*v*) CO_2(g)_ for 4.5 h.

Immediately prior to harvest, cell confluency (%) was measured using an IncuCyte Zoom^®^ live cell imaging system (Essen BioScience, Sydney, Australia). For this, 4 × 4 field HD phase-contrast images of each well were acquired using a 10x objective lens (Nikon, Rhodes, Australia) and analyses were completed by the IncuCyte Zoom^®^ system software with user input for segmentation adjustment (differentiation of background from cells). After SAA treatment of cells, the HPSS overlay was collected and stored at −80 °C for further analyses. Treated cells were harvested in buffer (50 mM phosphate buffered saline (PBS) containing: 1 mM EDTA, 10 μM butylated hydroxytoluene and 1X Complete™ protease cocktail inhibitor tablet (Roche, Bern, Switzerland) and then exposed to 3 freeze-thaw cycles. The lysate was collected and stored at −80 °C for subsequent analyses. For all biochemical assessments, total cell protein was determined using the bicinchoninic acid assay with bovine serum albumin used as standard.

### 6.4. Gene Analysis 

TF and TNFα were chosen for gene analysis as SAA is known to stimulate expression of these genes in a range of cell types. Where required, total mRNA was extracted from cell lysates using a commercial Isolate II RNA Mini Kit (Bioline, Sydney, Australia) and cDNA thereof was synthesised immediately using a Tetro cDNA Synthesis Kit (Bioline, Sydney, Australia) with an Eppendorf MasterCycler gradient System (Eppendorf, Sydney, Australia). Reverse-transcriptive polymerase chain reaction (RT-PCR) was performed with 0.5–6 μL cDNA (final volume was optimised by normalising the corresponding house-keeping gene). A PCR master mix was prepared by adding 1 μL of specific forward primer and 1 μL of reverse primer (Table 1) and 12.5 μL 2x MyTaq Red (Bioline, Sydney, Australia) and 14.5 μL of the PCR master mix was added to each PCR tube followed by cDNA and finally diluting to 20 μL with DEPC-treated water (Bioline, Sydney, Australia).

PCR reactions were cycled at 95 °C for 1 min to activate DNA polymerase, heated to denature step at 95 °C for 15 s, allowed to anneal at 60 °C for 15 s followed by elongation at 72 °C for 10 s. Amplified products were visualised using a 2% (*w*/*v*) agarose gel and SYBR^®^Safe (Life Technologies, Carlsbad, CA, USA). PCR reaction samples were loaded into wells with 6 μL of Hyperladder I (Bioline, Sydney, Australia) and PCR products separated using electrophoresis (Bio-Rad, Sydney, Australia) for 45 min at 90 V. The gel was imaged on a GelDoc™ MP system (Bio-Rad, Sydney, Australia) using ImageLab™ Software V5.2 (Bio-Rad, Sydney, Australia). Gene expression was quantified using gel densitometry with ImageLab™ Software in conjunction with ImageJ (National Institutes of Health, Bethesda, MD, USA). 

### 6.5. Sandwich Enzyme-Linked Immunosorbent Assay (ELISA)

Cells were normalized for cell density/confluency and interleukin-6 and ERK1/2 proteins were quantified in media samples and cell lysates (after normalising for total cell protein) and aortae homogenates using a commercial sandwich ELISA (Elisakit.com, Australia). As per manufacturers guidelines absorbance readings were recorded at 450 nm using an Infinite^®^ M200 PRO Plate reader (Tecan, Männedorf, Germany) and analysed using Microsoft Excel (2013, v7). Standard curves generated on the same plate were employed to express protein units in pg/mL. Where required all data was normalised to the corresponding level of cell confluence (expressed as a percentage %) determined using an IncuCyte system immediately prior to cell harvest for ELISA (See Appendix A).

### 6.6. Immunofluorescence 

Immunofluorescence imaging of HCtAEC was performed as described previously [23] for TNF and IL-6. Briefly, cells were grown to ~80% confluence on Nunc™ Lab-Tek™ II Chamber Slides (Thermo Scientific, Melbourne, Australia) before treatment with SAA (10 ug/uL) in the absence and presence of BAY-11 (10 uM). Slides were then washed 3x with HPSS and cells fixed with ice-cold acetone for 20 min before addition of rabbit anti-human primary antibody (Abcam, Sydney, Australia), (1:50 *v*/*v* dilution in antibody diluent (0.1% (*v*/*v*), Triton-X 100, 1% (*w*/*v*) BSA in TBS) and incubation at 4 °C overnight. The chambers were washed gently with TBS-T three times for 3 min and treated with a secondary IgG-Alexa Fluor^®^ 594 conjugated antibody (Life Technologies, Sydney, Australia) at 1:200 *v*/*v* final dilution for 45 min at 20 °C followed by 3 wash cycles (TBS-T).

A glass cover slide was mounted onto chamber slides using SlowFade Gold Antifade Reagent with DAPI (Life Technologies, USA). The slides were allowed to dry overnight in the dark at 4 °C and then sealed with clear nail varnish. Slides were imaged on a Zeiss Axio Scope A.1 (Carl Zeiss, Melbourne, Australia) using a 20x objective lens and Zen2 Lite software (Carl Zeiss, Melbourne, Australia) was utilised to obtain images. All images were converted to TIF and were manually edited by adjusting the brightness and contrast uniformly to ensure balanced contrast and to remove background distortions using Microsoft PowerPoint (Office v7, 2013).

### 6.7. Animal Studies

#### 6.7.1. Leukocyte Adhesion 

After gaining ethical approval from the local Animal Ethics Committee (E/1573/2015/B; 30 August 2015) (Melbourne, Australia), 8 weeks old male atherosclerosis-prone Apolipoprotein E^-^/^-^ mice on a C57BL/6 background were allowed to acclimatise for one week prior to experimentation. Intraperitoneal (*i.p*.) administration of BAY 11-7082 (5 mg/kg) or sterile PBS (as vehicle control) commenced on day 0 and was repeated on days 4, 7, 10 and 13. In parallel, recombinant SAA (120 μg/kg; Lonza Australia, Mount Waverly, Australia) was administered via *i.p*. injection on days 8, 11 and 13, while the same vehicle control mice (above) were administrated sterile PBS (100 μL, 0.01M; via *i.p*. injection) on days 0, 4, 7, 8, 10, 11 and 13 as a further control measure. All mice were anaesthetised and culled via cervical dislocation on day 14 for use in *ex vivo* vascular adhesion studies as described previously [62].

Leukocyte-endothelial interactions were observed using an *ex vivo* dynamic flow adhesion assay [63]. Briefly, abdominal aortae were excised and aortic surfaces thoroughly cleaned of excess fat to facilitate clear imaging. Each end of the abdominal aorta was then carefully mounted on the cannula in a vessel chamber. Whole human blood was freshly isolated on the day of the experiment and labelled with VybrantDil dye (Invitrogen, Carlsbad, CA, USA) (final dilution 1:1000 *v*/*v*) for 10 min and then perfused at a rate of 7.5 mL/h through the isolated aorta and leukocyte adhesion was visualized by fluorescence microscope. Images were acquired at 160x magnification with a 1.4 mm field and readings were taken at 1 field along the vessel at 5 separate time points in 2.5 min intervals for 30 s.

#### 6.7.2. Vascular Function Studies

Next, we extended our analysis to assess the vascular functionality (Srague-Dawley) using freshly isolated rat aortic rendered surplus and made available to other researchers in line with animal ethics guidelines vessels (Sydney South West Area Health Services ethics protocols 2014/020 and 2014/030), thereby minimising the usage of new animals. Briefly, isolated aortae were excised and flushed with HBSS and vascular activity study was performed within 12–16 h of harvest. Aortae were then placed in modified Krebs-Henseleit solution (containing in mM: 11 D-glucose, 1.2 MgSO_4_, 12 KH_2_PO_4_, 4.7 KCl, 120 NaCl, 25 NaHCO_3_ and 2.5 CaCl_2_·2H_2_O) and cut into 3 mm ring segments as described previously [2]. 

Aortic rings were then mounted in a Myobath system (World Precision Instruments, Sarasota, FL, USA) containing 15 mL of modified Krebs–Henseleit solution aerated at 37 °C with carbogen gas (5% *v*/*v* CO_2_/oxygen). Next, individual rings were contracted with phenylephrine (10^−7^–10^−5^ mol/L) and the dose that elicited half-maximal contraction force in each mounted aortic segment was selected for further studies. Mounted rings were then treated with vehicle (control) or SAA (final concentration 1 μg/mL a dose known to activate the endothelium and impair vaso-relaxation [13] for 4 h followed by thorough washing prior to relaxation assessment. In some experiments, the aortic ring segments were pre-treated with BAY11-7082 (10 μM) for 1 h prior to addition of SAA. Concentration–response curves (10^−6^–10^−2^ mol/L) were then generated for endothelium-dependent (acetylcholine; ACh) or endothelium-independent (sodium nitroprusside; SNP) relaxation in the presence of added indomethacin (25 μM). Vessel relaxation was expressed as a percentage of initial phenylephrine-mediated contraction force. At the completion of the experiments each vessel was immediately frozen in liquid nitrogen and stored at −80 °C for further biochemical assessment.

### 6.8. Measurement of Aortic cGMP

Aortic levels of the vaso-active effector molecule cGMP were determined using a commercial ELISA kit (Sapphire Biosciences, Redfern, Sydney, Australia). Where required, aorta samples were thawed, homogenized and centrifuged (16,000× *g*, 10 min, 4 °C) to obtain cytosolic protein from the supernatant fraction, as previously described [26]. Aortic cGMP levels were subsequently determined by ELISA. Standard curves generated on the same plate were employed to express protein units in pg/mL; all data was normalized to total aortic homogenate protein.

### 6.9. Statistical Analysis

Statistical analysis was performed using the GraphPad^®^ Prism software Version 6.0 (GraphPad Software Inc., La Jolla, CA, USA). Data was analysed using one-way analysis of variance (ANOVA) with Tukey’s multiple comparisons test to compare differences between groups. Statistical significance was reached at 95% confidence interval (*p* < 0.05).

## Figures and Tables

**Figure 1 ijms-20-00105-f001:**
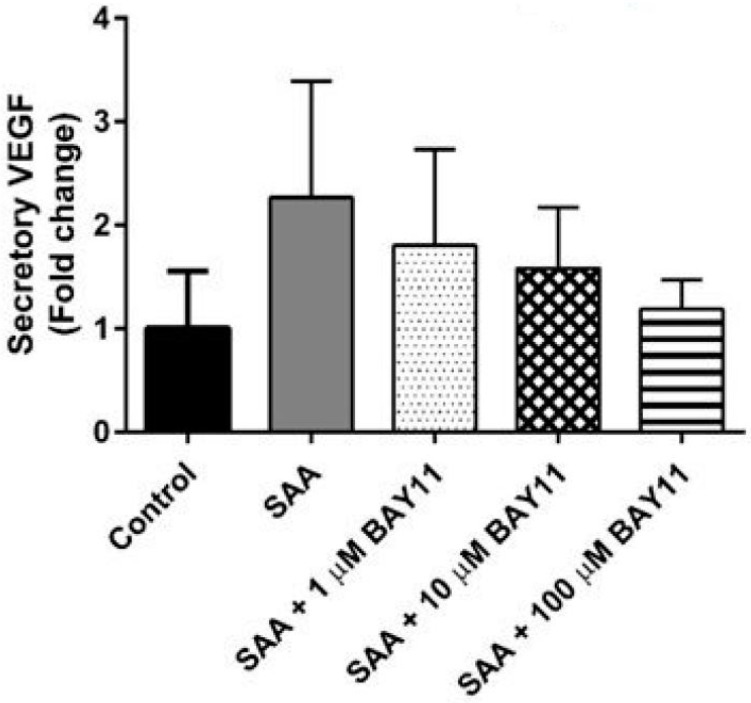
Secretion of VEGF from HCtAEC in response to SAA in the absence and presence of the NFκB inhibitor BAY11-7082. HCtAEC were pre-incubated (1.5 h) with 0, 1, 10 or 100 μM BAY11-7082 or vehicle (control) then treated with 10 μg/mL SAA. After 4.5 h the levels of secretory VEGF in cell supernatants was assessed by commercial ELISA assays. Data were expressed as a fold-change compared to control cells. Data represents the mean ± SD (n = 4 individual experiments, each performed in duplicate; therefore, data represent 4 technical replicates). * *p* < 0.05 compared to control cells. All data was normalised to the corresponding level of cell confluence (expressed as a percentage %) determined using an IncuCyte system immediately prior to cell harvest for ELISA. This approach to normalizing data was necessary as there was some level of toxicity for BAY11-7082 particularly at the higher doses of inhibitor tested.

**Figure 2 ijms-20-00105-f002:**
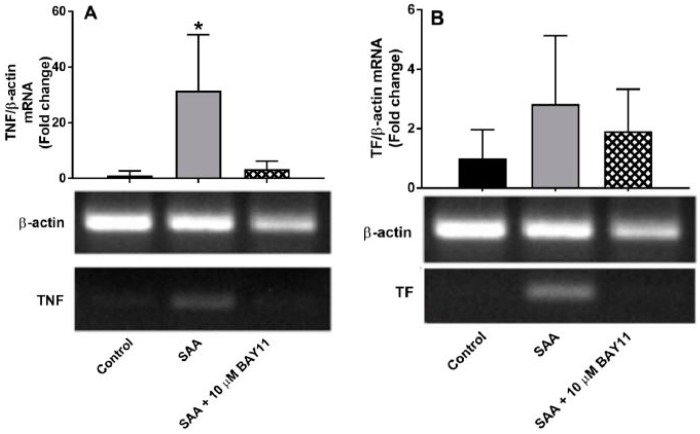
Relative gene expression in HCtAEC in response to SAA in the absence and presence BAY11-7082 under normoglycaemic conditions. HCtAEC were incubated with 10 μg/mL SAA or vehicle (control) with or without 10 μM BAY11-7082 pre-treatment under normoglycaemic conditions (5 mM glucose). Total RNA was isolated after 4.5 h and gene regulation was assessed. Representative gels and semi-quantitative analysis are shown for (**A**) TNF (**B**) TF. Gene expression was normalised against β-actin and results represent fold change compared to control cells. Data shows the mean ± SD (n = 3 individual experiments each completed in duplicate; therefore, data represent 3 technical replicates); * *p* < 0.05 compared to control, where possible, target gene products were confirmed by sequence analysis (refer to Appendix A).

**Figure 3 ijms-20-00105-f003:**
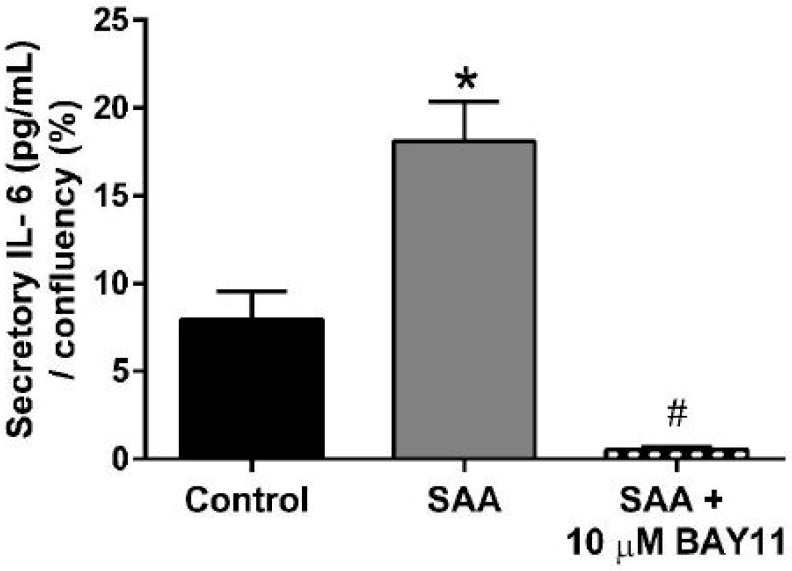
BAY11-7082 reduces secretory IL-6 induced by SAA. HCtAEC were treated with SAA and BAY11-7082 as defined in methods. A commercial ELISA kit was used to quantify levels of secretory IL-6 in the cell supernatant. All data was normalised to the corresponding level of cell confluence (expressed as a percentage %) determined using an IncuCyte system immediately prior to cell harvest for ELISA (See Appendix A). Data represent the mean ± SD (n = 6 samples, each completed in duplicate; therefore, data represent 3 technical replicates). * Different to the control; *p* < 0.0001. # Different to SAA-treated cells; *p* < 0.0001.

**Figure 4 ijms-20-00105-f004:**
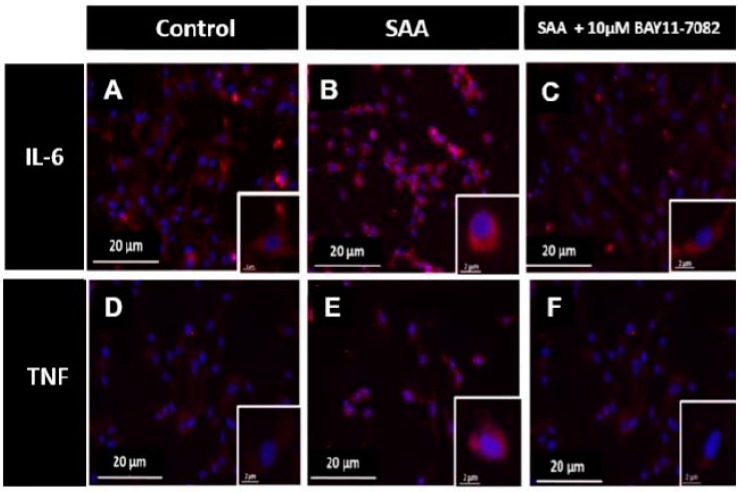
Changes in intracellular IL-6 and TNF in response to SAA and BAY-11-7082. HCtAEC were treated with SAA and with and without BAY11-7082 pre-incubation. Next, cellular TNF and IL-6 content and distribution was visualised by immunocytochemistry. For all images shown, nuclei were stained with DAPI (blue) or IL-6^+^ (**A**–**C**) or TNF^+^ (**D**–**F**) immune-fluorescence using an appropriate secondary antibody (red) under 20× of fluorescence microscope. Representative images are shown for cells treated with DMSO (vehicle control) (**A**,**D**), 10 μg/mL SAA (**B**,**E**) or pre-incubated with 10 μM BAY11-7082 (**C**,**F**) prior to addition of 10 μg/mL SAA. Data shown are representative of 5 fields and (n = 2 control, n = 2 SAA or n = 3 BAY11-7082 + SAA independent repeat studies). Insets to panels **A**–**F** show higher magnification images of representative single cells from the different treatment groups.

**Figure 5 ijms-20-00105-f005:**
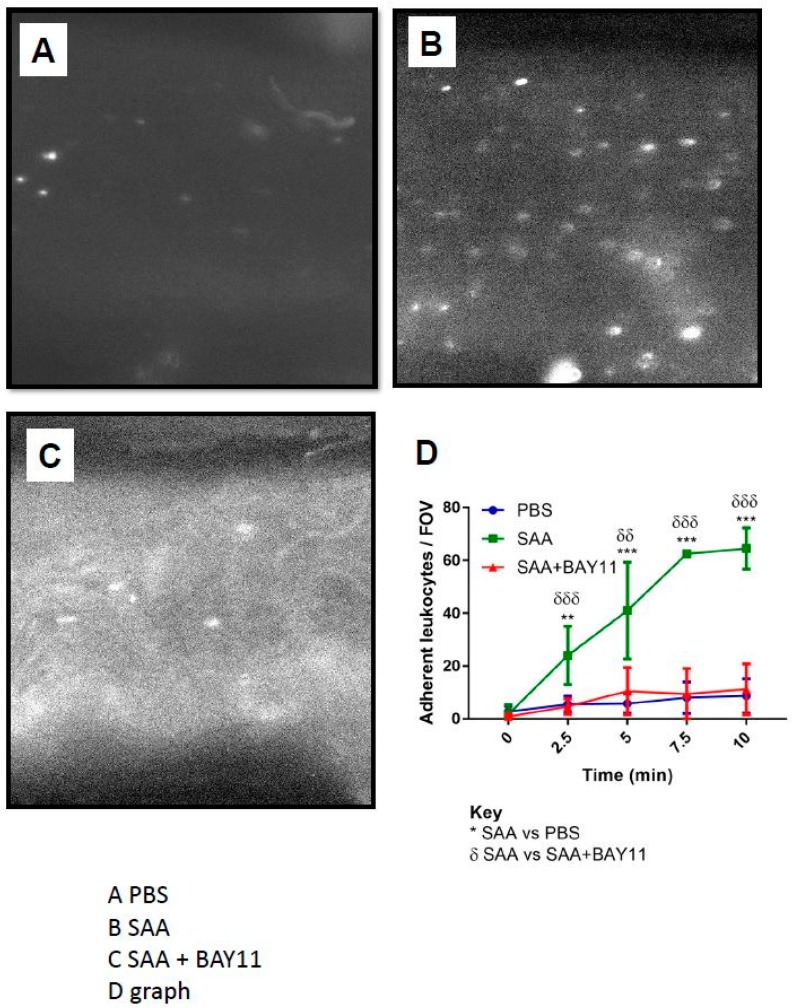
BAY11-7082 reduces *ex vivo* leukocyte adhesion induced by SAA in mice aorta. Mice were administered BAY11-7082 (5 mg/kg) or PBS (control) over a 14day period in parallel with administration of SAA (120 μg/kg; PBS as control) as described in the Methods Section. Aorta were subsequently isolated and subjected to ex vivo leukocyte adhesion studies using total labelled leukocytes. Arrows indicate leukocytes adhered to the aortic surface of mice treated with (**A**) SAA alone and (**B**) SAA + BAY11-7082. (**C**) Adherent leukocytes were quantified per field of view (FOV) by counting the number of dye-stained leukocytes that ceased motion during a 30 s recording at various time intervals under 160x magnification. Data represent mean ± SD. Independent aortic samples n = 4 SAA (time = 0; 2.5 min); n = 3 SAA (time = 5 min); n = 2 SAA (time = 7.5; 10 min); n = 6 SAA + BAY11-7082 (time = 0; 2.5; 5 min); n = 5 SAA + BAY11-7082 (time = 0 = 7.5; 10 min). * Different to SAA-only group ** *p* < 0.05 and *** *p* < 0.001. Data represent 3 technical and 3 biological replicates.

**Figure 6 ijms-20-00105-f006:**
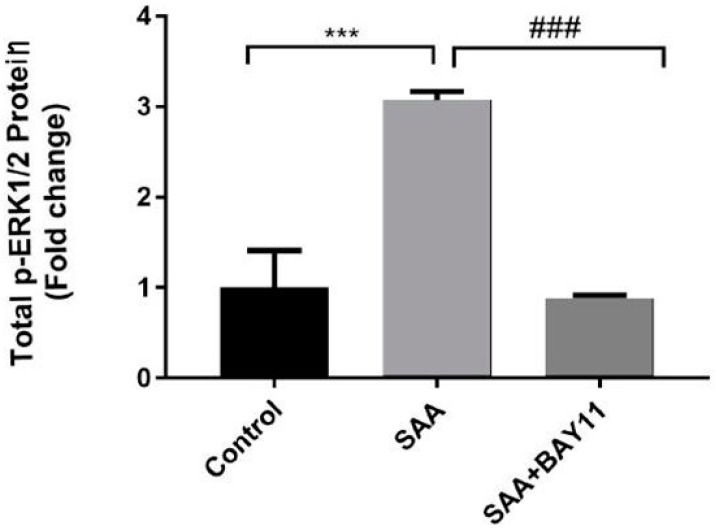
ERK1/2 levels in aortae homogenates. Total phosphorylated ERK1/2 (p-ERK1/2) protein was analysed in homogenates prepared from isolated mouse aortae (refer to Methods section) using the commercial ERK1/2 (pT202/Y204) SimpleStep ELISA™ Kit (Abcam, Cambridge, MA, USA) (Catalogue number: ab176640) as per manufacturers guidelines. Absorbance readings were recorded at 450 nm using an Infinite^®^ M200 PRO Plate reader (Tecan, Männedorf, Germany) and analysed using Microsoft Excel (2013, v7). Standard curves generated on the same plate were employed to express protein units in ug/mL. Homogenate samples tested were from control (n = 4), SAA (n = 2) and SAA + BAY11 (n = 4), each measured in duplicate. *** *p* < 0.0001 and ### *p* < 0.0001.

**Figure 7 ijms-20-00105-f007:**
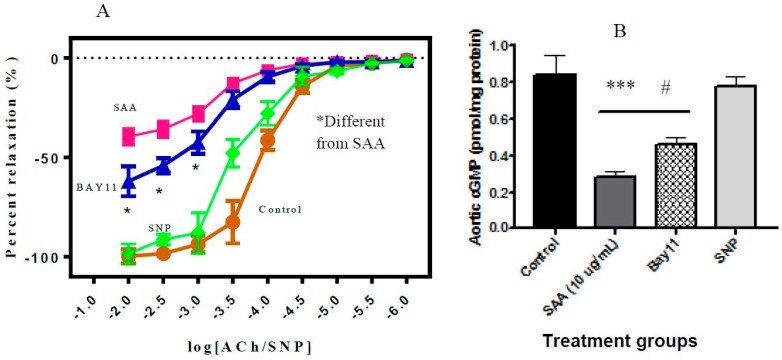
SAA endothelium-dependent vaso-relaxation is partially reversed by BAY11-7082. Rat aorta were isolated, cut into ring segments and mounted onto a force transducer. Rings were incubated with 10 μg SAA protein/mL (■), pre-treated with BAY11 (▲), or control (●) and then pre-constricted with phenylephrine as described in the Methods section. (**A**) Vascular relaxation was then assessed in response to the endothelium-dependent (acetylcholine [ACh]) or endothelium-independent (sodium nitroprusside [SNP], positive control (♦) stimuli. * Different to the rings treated with SAA in the absence of BAY11-7082; *p* < 0.01. (**B**) Levels of cGMP were determined with a commercial ELISA assay. Data represent mean ± SD (SAA: n = 5; BAY11: n = 5; Control: n = 4; SNP: n = 4) independent aortic samples. Different to the control endothelium-dependent relaxation; # *p* < 0.05. Different to rat aortic rings treated with SAA in the absence of Bay 11-7082; *** *p* < 0.001.

**Table 1 ijms-20-00105-t001:** Primer sequences employed in this study ^a.^

Primer	Forward Sequence (5′-3′)	Reverse Sequence (5′-3′)	Accession #
**β-actin**	AGCACTGTGTTGGCGTACAG	GGACTTCGAGCAAGAGATGG	XM_006715764.1
**TNF-α**	CAGAGGGCCTGTACCTCATC	GGAAGACCCCTCCCAGATAG	NM_000594.3
**TF**	GTGACCTCACCGACGAGATT	CCGAGGTTTGTCTCCAGGTA	NM_001178096.1
**NFκB**	TGGAAGCACGAATGACAGA	TGAGGTCCATCTCCTTGGTC	NM_001319226.1

^a^ Primer sequences used in RT-PCR studies with annealing temperature set to 60 °C for all experiments. Primers were synthesised by Sigma-Aldrich, Australia and Geneworks, Australia. β-actin used as a housekeeping gene to normalise results. All primer sequences were verified as targeting the gene of interest in the species nominated using Blast Search (an on-line database screening freely available at the NIH site: https://blast.ncbi.nlm.nih.gov/Blast.cgi?PAGE_TYPE=BlastSearch) and validated by gene sequencing (refer to Appendix A); identifying accessions numbers are as reported above.

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
