# Peer review of "NFκB Inhibition Mitigates Serum Amyloid A-Induced Pro-Atherogenic Responses in Endothelial Cells and Leukocyte Adhesion and Adverse Changes to Endothelium Function in Isolated Aorta"

_ijms, 2018, doi:10.3390/ijms20010105_

Round 1
Reviewer 1 Report
Vallejo et al submit a paper entitled "NFkB inhibition mitigates serum amyloid A-induced pro-atherogenic responses in endothelial cells and leukocyte adhesion and adverse changes to endothelium function in isolated aorta". They show using a combination of in vitro and in vivo experiments that Serum Amyloid A (SAA) increases the expression of IL6 and TNF alpha in human carotid endothelial cells. . This expression was inhibited by BAY11-7082, a pharmacologic inhibitor of NFKB. BAY11- 28 treatment also significantly decreased SAA-mediated leukocyte adhesion to apolipoprotein E-deficient mouse aorta in ex vivo . The experiments are interesting, and novel. The paper is well written.
Major comments.
Why did the authors not detect IL6 expression in figure 2 ?
Why did the authors detect only IL6 expression in figure 3, and not TNF alpha expression?
The authors should indicate clearly in all figures the exact number of technical and biological repilcats performed.
minor:
Cell culture: what passages of the cells were used for the experiments?
In Mat and Meth, ELISA, the authors justify the fact that they quantify IL6. these points are valid but should appear in discussion rather than in Mat and Meth
Can the authors justify the use of beta-actin as gene control in RT-qPCR experiments? Why not choose a gene specific for endothelial cells ?
In RT-qPCR experiments, did the authors check for RNA degradation after extraction?
page 5 line 23 please correct "vascular endothelium"
Author Response
Dear Reviewers,
We thank you very much for your constructive comments and feedback on our manuscript titled “NFκB inhibition mitigates serum amyloid A-induced pro-atherogenic responses in endothelial cells and leukocyte adhesion and adverse changes to endothelium function in isolated aorta”. We have addressed the comments and where relevant added the information in the main document including additional new data where applicable.
REVIEWER 1.
Major comments:
1. Why did the authors not detect IL6 expression in figure 2?
Based on our previous work with SAA stimulation of other inflammatory cell types we expected a higher gene expression of TNF alpha which ultimately would initiate IL-6 protein expression therefore, TNF alpha was our primary target. Completely consistent with this notion, others have demonstrated that TNF alpha has elicited expression of IL-6 at gene and protein levels in rat intestinal epithelial cells (Mv Gee et al., Immunology. 1995 Sep; 86(1): 6–11). Therefore, we focused on measurements of protein levels of TNF and IL-6 (immune fluorescent microscopy; Figure 4 in original and revised manuscripts) and gene expression of TNF alpha.
The following text has been added to page 4 bottom in the revised manuscript:
Enhanced expression of TNF is linked to the downstream production of secretory IL-6 as marker of bioactivity for this cytokine. For example, TNF alpha has been demonstrated to elicited expression of IL-6 at gene and protein levels in rat intestinal epithelial cells (Mv Gee et al., Immunology. 1995 Sep; 86(1): 6–11). In our hands, cell confluency-normalised secretory IL-6 protein increased (~10-fold; P<0.05) in HCtAEC exposed to 10 μg/mL of pro-inflammatory SAA.
2. Why did the authors detect only IL6 expression in figure 3, and not TNF alpha expression?
The cytokine TNF alpha is linked to a range of downstream inflammatory mediators such as interlukins. Given the established link between TNF and IL-6 (see response to Q1 above) we decided to measure the gene expression of TNF alpha, which was significantly over expressed in SAA induced cells and BAY11-7082 showed significant inhibition as demonstrated in Figure 2, and then link this axis of cytokine changes to IL-6 measured by commercial ELISA (Figure 3), which responded as anticipated by the changes in TNF gene expression. Furthermore, we have assessed both IL-6 and TNF protein the cultured cells in Figure 4 (as indicated above).
3. The authors should indicate clearly in all figures the exact number of technical and biological replicates. performed.
All experiments are representative of minimum 3 technical and 3 biological replicates. The relevant information is now added in the legends of each corresponding figure in the revised manuscript as requested by the reviewer.
Minor comments:
1. Cell culture: what passages of the cells were used for the experiments?
We routinely performed assays with early passage cells. Thus, each in vitro test was performed on passage 4 of the cell lines tested. The information is added in the material and methods section page no 11 under the subheading “cell culture”.
2. In Mat and Meth, ELISA, the authors justify the fact that they quantify IL6. These points are valid but should appear in discussion rather than in Mat and Meth
As suggested by the reviewer, the specified information has moved to the discussion section now appearing on page 8, centre of paragraph 2 of the revised manuscript.
3. Can the authors justify the use of beta-actin as gene control in RT-qPCR experiments? Why not choose a gene specific for endothelial cells?
The selection of a suitable house-keeping gene for assessment of changes in mRNA expression requires (i) stable expression and (ii) relatively inert expression in the presence of external stimuli. We initially selected beta actin and GAPDH as the preferred house-keeping candidates and determined that beta actin gave optimal response compared with GAPDH, probably due to the fact that GAPDH can be sensitive to altered cellular redox status and we have demonstrated that SAA elicits the production of reactive oxygen species in endothelial cells (Witting PK et al., Free Radic Biol Med. 2011;51(7):1390-8). Therefore, beta actin was selected as the preferred house-keeping after validation gene sequencing (refer to supplementary data for validating gene sequencing of RT-PCR assessment for beta actin).
4. RT-qPCR experiments, did the authors check for RNA degradation after extraction?
All extractions steps were performed on ice, and mRNA isolated after completion of the relevant experiments were immediately converted to cDNA and carefully stored at low temperature to avoid any degradation as per supplier instructions (Isolate II miRNA kit, Bioline, Sydney, Australia). To further validate the suitability of cDNA for assessing gene regulation all gene products assessed by RT-PCR were reconfirmed by gene sequencing (refer to gene sequencing data in the supplementary section).
5. page 5 line 23 please correct "vascular endothelium
The terminology has been corrected.
Reviewer 2 Report
In this manuscript, the authors assess whether the NF-kB inhibitor BAY11-7082 reduces pro-inflammatory responses of endothelial cells to serum amyloid A. The results shown are generally clear, however there are concerns about the physiological relevance of this work, and there are also some problems with experimental design. Major and minor comments are detailed below:
Major Comments:
1. There is some question whether recombinant SAA treatment appropriately models the physiological effects of SAA overexpression in vivo. This is, in part, due to the majority of SAA in the circulation being bound to HDL. Recent works have revealed that pro-inflammatory functions of SAA are not observed when SAA is HDL-bound (Kim et al. Cytokine. 2013 Feb;61(2):506-12). The same study notes that the recombinant human form of SAA that has been widely used to study SAA pro-inflammatory activity is a hybrid molecule that contains an N-terminal methionine and two amino acid base pair substitutions. It is unclear whether this version of SAA functions similarly to natural sequences of human SAA1 or SAA2 isoforms. The degree to which endogenous SAA activates pro-inflammatory signaling in vivo could be minimal.
2. In the supplement, the cell confluency is about half when cells are treated with the NF-kB inhibitor. Is this a cytotoxic dose? Although normalization is done to correct for differences in cell numbers with treatment, this makes interpretation of the data more difficult.
3. Significant differences are not indicated in figure 1. It is also unclear why VEGF was used to establish the dose response of BAY11-7082- despite its association with atherosclerosis, this was not investigated mechanistically in the remainder of the manuscript.
4. Nuclear localization or activity of NF-KB in response to SAA should be shown in the cell model used, ideally with the dose of BAY11-7082 used for all studies.
5. It appears that in Figure 2, endpoint RT-PCR was used to quantify mRNA. Endpoint PCR is a non-quantitative method; qRT-PCR using either the delta CT method or absolute mRNA quantification should be done instead. Quantification of secreted TNF protein as also acceptable.
6. Figure 5 lacks control groups not receiving SAA treatment. It’s important to show statistical comparisons among SAA-untreated groups with or without BAY11-7082 to demonstrate drug efficacy in vivo. It appears that this was done in results, but the data should be shown and included in analysis.
7. It would be helpful to expand on the experiments done in Figure 5 and determine whether isolated aortas show differences in cytokine expression that parallel results in the cell line.
8. Some background should be provided on the validity of using human leukocytes with mouse endothelial cells for leukocyte adhesion assays.
Minor comments
1. The methods have a good deal of background information/justification that may be more appropriately included in either the intro or results sections.
2. The recombinant form of SAA used is not specified in methods; Preprotech makes two versions, and so the product number should be specified for clarity.
3. The legend for Figure 6 does not clearly describe the groups. Including a legend for Figure 6A would be helpful. Figure 6 B labeling is also confusing, and appears to be single treatments as shown.
Author Response
Dear Reviewers,
We thank you very much for your constructive comments and feedback on our manuscript titled “NFκB inhibition mitigates serum amyloid A-induced pro-atherogenic responses in endothelial cells and leukocyte adhesion and adverse changes to endothelium function in isolated aorta”. We have addressed the comments and where relevant added the information in the main document including additional new data where applicable.
1. There is some question whether recombinant SAA treatment appropriately models the physiological effects of SAA overexpression in vivo. This is, in part, due to the majority of SAA in the circulation being bound to HDL. Recent works have revealed that pro-inflammatory functions of SAA are not observed when SAA is HDL-bound (Kim et al. Cytokine. 2013 Feb;61(2):506-12). The same study notes that the recombinant human form of SAA that has been widely used to study SAA pro-inflammatory activity is a hybrid molecule that contains an N-terminal methionine and two amino acid base pair substitutions. It is unclear whether this version of SAA functions similarly to natural sequences of human SAA1 or SAA2 isoforms. The degree to which endogenous SAA activates pro-inflammatory signaling in vivo could be minimal.
We completely agree on this controversy as to the bioactivity of SAA in the presence of HDL and have added following information in our limitation section (page 10) to address this issue. Overall, we deduce that the ratio of SAA-to-HDL may be crucial in defining SAA bioactivity in the presence of this lipoprotein.
“One clear limitation to this study is the mode of delivery of SAA to the endothelial cells in the absence of serum. SAA is an apolipoprotein for HDL and avidly binds to this lipoprotein and in the process displace apolipoprotein AI [56, 57]. SAA bioactivity may be regulated by binding to HDL. We have previously shown that this binding of SAA to HDL becomes saturated at approximately 5 mol SAA per mol HDL [58]. Therefore, for SAA to elicit a biological response on endothelial cells levels of SAA should be in significant excess to circulating HDL (for example, when the mol ratio SAA:HDL is > 5). Such conditions may be evident for chronic inflammatory disorders such as diabetes and rheumatoid arthritis [59, 60] where SAA levels are chronically elevated and natural variation amongst humans can see lower levels of HDL that predispose to vascular inflammation [61] and thereby, a high SAA to HDL ratio may compromise HDL anti-inflammatory activity to promote atherosclerosis [62]”.
With reference to the bioactivity of recombinant SAA, we note that others have suggested that the subtle changes in the protein sequence may render recombinant SAA in active in vivo and that this controversy continues with a recent review article of the field pointing to yet other research that indicates recombinant SAA retains similar pro-inflammatory activity to mammalian SAA [refer to Sack GH Jr. Mol Med. 2018 Aug 30;24(1):46.]. We also note that we routinely and exhaustively test preparations of recombinant SAA for LPS contamination and determined that batches employed here contained <2 pg LPS/µg SAA/mL. This low-level contamination was unable to induce pro-inflammatory/pro-coagulant responses in human peripheral blood monocytes that are highly sensitive to LPS and denaturing the SAA protein by heating to 100 °C ablates it biological activity on isolated peripheral blood monocytes while similar treatment of LPS has no impact on its biological action. This information is cited in the Materials and Methods section (page 11; centre of paragraph 2 of the revised manuscript).
2. In the supplement, the cell confluency is about half when cells are treated with the NF-kB inhibitor. Is this a cytotoxic dose? Although normalization is done to correct for differences in cell numbers with treatment, this makes interpretation of the data more difficult.
All data was normalised to the corresponding level of cell confluence (expressed as a percentage %) determined using an IncuCyte system immediately prior to cell harvest for ELISA. This approach to normalizing data was necessary as there was some level of toxicity for BAY11-7082 particularly at the higher doses of inhibitor tested (refer to Figure 1 legends). The selected dose (10 μM) of BAY11-7082 was validated as being optimal for these cell studies based on the sufficient inhibitory effect on secretary VEGF. The selected dose is lower than used by others as mentioned in the manuscript (Lappas et al., 2005: Endocrinology, 2005. 146(3): p. 1491-7, Zhang et al., 2017: J Pain Res, 2017. 10: p. 375-382) and provides an advantage of minimal non-specific binding and lower toxicity profile.
3. Significant differences are not indicated in figure 1. It is also unclear why VEGF was used to establish the dose response of BAY11-7082- despite its association with atherosclerosis, this was not investigated mechanistically in the remainder of the manuscript.
We employed VEGF as a useful downstream biomarker of SAA activity as we have demonstrated previously (Cai X et a., Clin Exp Pharmacol Physiol. 2013;40(9):662-70). As indicated in the original manuscript we did not determine a statistical significance for the diminution of VEGF protein expression as a function of BAY11-7082 dose, Figure 1. However, the VEGF secretary levels were nearly 40% reduced by 10 μM dose of BAY11-7082 and this dose showed significant changes in gene and protein expression of TNF and IL-6 respectively (Figure 2 and 3), the two main inflammatory key players. The possible lack of significance in figure one could be the number of technical replicates (n=3) which might have impacted the statistical outcome, nevertheless the dosage selected was active in our other investigations.
We did not consider further assessment of VEGF in the context of SAA-mediated inflammation as this biomarker is associated with endothelial cell migration/proliferation, which was not the focus of this study. Instead VEGF was merely a biomarker that was readily monitored in the presence of SAA with or without the added pharmacologic inhibitor BAY11-7082.
3. Nuclear localization or activity of NF-KB in response to SAA should be shown in the cell model used, ideally with the dose of BAY11-7082 used for all studies.
We admit that this is a limitation of the current study. However, we selected the proteins TNF and IL-6 as they form part of the NfKB axis of activity with NfKB activation leading to TNF production and downstream IL-6 expression (as demonstrated by immune-fluorescence microscopy for the latter two proteins) thereby inferring NFkB activation.
4. It appears that in Figure 2, endpoint RT-PCR was used to quantify mRNA. Endpoint PCR is a non-quantitative method; qRT-PCR using either the delta CT method or absolute mRNA quantification should be done instead. Quantification of secreted TNF protein as also acceptable.
The reviewer correctly points out that endpoint RT-PCR can suffer from saturation and is limited as a quantitative procedure. However, this method is useful to compare gross changes in mRNA levels when strongly elicited by a pro-inflammatory mediator such as SAA. Furthermore, we are confident the gene products investigated were correct as judged by gene sequencing analysis to reconfirm the validity of TNF/TF (refer to supplementary data). In addition, we have demonstrated that the gene regulation identified here does progress to altered protein expression of a downstream mediator known to be regulated by TNF. Therefore, the linkage of these data sets validates the approach and offsets the argument that quantitative gene analyses is absolutely necessary.
5. Figure 5 lacks control groups not receiving SAA treatment. It’s important to show statistical comparisons among SAA-untreated groups with or without BAY11-7082 to demonstrate drug efficacy in vivo. It appears that this was done in results, but the data should be shown and included in analysis.
As suggested by the reviewer, we have now included the PBS (control) data along with SAA and SAA+BAY11-7082 groups in an updated Figure 5 in the revised manuscript.
6. It would be helpful to expand on the experiments done in Figure 5 and determine whether isolated aortas show differences in cytokine expression that parallel results in the cell line.
We have utilised the notion that the TNF/NfKB axis of activity extends beyond IL-6 (see response to Reviewer 2; Question 3) and also involves activation of MAPKs such as ERK1/2 in endothelial cells; this approach is validated by the available literature (refer to new references [Zhong, X., et al., Biochem Biophys Res Commun. 2012:24;425(2):401-6; Luo, S.F et al., Arthritis Rheum. 2010 Jan;62(1):105-16.; Ho, A.W et al., Immunobiology. 2008;213(7):533-44) in the revised manuscript. Thus, we have analysed homogenates of isolated aortae from ApoE mice in all treatment groups and the new data demonstrate that SAA significantly activates ERK1/2 whereas, the pharmacological inhibitor BAY11-7082 inhibits ERK1/2 activation yielding similar levels as the control. Together, these new data suggest that SAA induces NFkB activation not only in cultured endothelial cells but in intact aortic tissue where endothelial activation is shown to promote leukocyte adhesion.
The relevant information on ERK1/2 activation in mouse aortae is added to the results (page 5 under leukocyte endothelial adhesion subheading) and discussion sections (page 7 bottom of third paragraph) in the revised manuscript
7. Some background should be provided on the validity of using human leukocytes with mouse endothelial cells for leukocyte adhesion assays.
We used human leukocytes in line with the established protocol (Michell et al., J Vis Exp. 2011 Aug 23;(54): doi: 10.3791/3221). Human leukocytes are used in these experiments owing to larger blood volume (50mL) daily required for the experiments. If mice were to be chosen, this would require about 70 mice per day (or 210 for the entire 3 days of experiments) to complete the experiments. Thus, to minimise the usage of animals, human leukocytes were used.
Minor comments
1. The methods have a good deal of background information/justification that may be more appropriately included in either the intro or results sections.
The relevant information on gene/protein analysis and corresponding justification has been rearranged in results (ref to page 4 paragraph 2 under gene expression heading) and discussion (ref to page 8 centre of paragraph 1) sections.
2. The recombinant form of SAA used is not specified in methods; Preprotech makes two versions, and so the product number should be specified for clarity.
We used recombinant Human Apo-SAA from Peprotech. Catalog Number: 300-13. The information is updated in the materials and method section (ref to page 11 paragraph 2).
3. The legend for Figure 6 does not clearly describe the groups. Including a legend for Figure 6A would be helpful. Figure 6 B labelling is also confusing, and appears to be single treatments as shown.
As suggested by the reviewer we have added the legends in the former Figure 6A (now numbered as 7A in revised manuscript) and revised the legends of former Figure 6B (now numbered 7B in revised manuscript).
Round 2
Reviewer 1 Report
Changes are satisfactory
Reviewer 2 Report
The reviewers have adequately responded to all comments, and only a couple minor issues are noted:
1. P-values should be specified in fig 6, and the single asterisk in fig 7 should be three asterisks for consistency.
2. In results it is stated that p-ERK1/2 levels increase, but it is not clear in Fig 6 what is actually quantified- is this phospho-ERK1/2, or the ratio of phospho/total?
Author Response
Dear Reviewers,
We thank you very much for your constructive comments and feedback on our manuscript titled “NFκB inhibition mitigates serum amyloid A-induced pro-atherogenic responses in endothelial cells and leukocyte adhesion and adverse changes to endothelium function in isolated aorta”. We have addressed the comments and where relevant added and highlighted the information in the main document.
REVIEWER 2.
Major comments:
1. P-values should be specified in fig 6, and the single asterisk in fig 7 should be three asterisks for consistency.
We have added the p value (p<0.0001) in the legends of figure 5 and as suggested by the reviewer added the asterisks in the figure 7B and corresponding legends.
2. In results it is stated that p-ERK1/2 levels increase, but it is not clear in Fig 6 what is actually quantified- is this phospho-ERK1/2, or the ratio of phospho/total?
Thank you for pointing out this missing information. We measured total p-ERK1/2 protein by ELISA as per instruction of the manufacturer as given in materials and methods section. The revised information is added in the figure 5 and corresponding legends and also in the results (ref to page 6 first line of first paragraph).